# NMDARs, Coincidence Detectors of Astrocytic and Neuronal Activities

**DOI:** 10.3390/ijms22147258

**Published:** 2021-07-06

**Authors:** Mark W. Sherwood, Stéphane H. R. Oliet, Aude Panatier

**Affiliations:** University of Bordeaux, INSERM, Neurocentre Magendie, U1215, F-3300 Bordeaux, France; stephane.oliet@inserm.fr

**Keywords:** astrocyte, coincident detection, d-serine, gliotransmission, glycine, neuron, NMDAR, trip-partite synapse, synapse cluster

## Abstract

Synaptic plasticity is an extensively studied cellular correlate of learning and memory in which NMDARs play a starring role. One of the most interesting features of NMDARs is their ability to act as a co-incident detector. It is unique amongst neurotransmitter receptors in this respect. Co-incident detection is possible because the opening of NMDARs requires membrane depolarisation and the binding of glutamate. Opening of NMDARs also requires a co-agonist. Although the dynamic regulation of glutamate and membrane depolarization have been well studied in coincident detection, the role of the co-agonist site is unexplored. It turns out that non-neuronal glial cells, astrocytes, regulate co-agonist availability, giving them the ability to influence synaptic plasticity. The unique morphology and spatial arrangement of astrocytes at the synaptic level affords them the capacity to sample and integrate information originating from unrelated synapses, regardless of any pre-synaptic and post-synaptic commonality. As astrocytes are classically considered slow responders, their influence at the synapse is widely recognized as modulatory. The aim herein is to reconsider the potential of astrocytes to participate directly in ongoing synaptic NMDAR activity and co-incident detection.

## 1. Introduction

Learning and memory are critical processes that define the temporal dimensions of our mental organization and determine our behaviour. Memory requires alterations in the brain and the ability to be persistently modified in response to specific neuronal activity. While there are several candidates, the most popular candidate is the synapse. Donald Hebb first hypothesized that synapses between neurons would be strengthened if they showed coincident activity [1]. This hypothesis has the potential to explain how associations between temporally linked events are formed. In 1973, Bliss and Lomo discovered long-term synaptic potentiation (LTP), a cellular mechanism of Hebbian plasticity which corresponds to a long-lasting potentiation of the synaptic strength [2,3]. Over the last half-century, substantial effort has been invested in understanding the molecular mechanisms of coincidence detection in synaptic plasticity. NMDARs materialized as critical molecules in LTP, learning and memory [4] are now appreciated for their crucial role in coincidence detection and the ability to transform specific activity patterns into long-lasting changes in synapses.

Classically, postsynaptic NMDARs detect coincident presynaptic and postsynaptic activities through glutamate binding and membrane depolarization. For NMDARs to open, co-agonist binding is also required. However, co-agonism was believed to be permissive; thus, its role in NMDAR function was overlooked for a long time [5,6,7], and as a result, the physiological significance is poorly understood. Nevertheless, recent evidence indicates that astrocytes play a central role in co-agonism and may participate in the tonic and active release of co-agonists [8,9,10,11,12,13,14]. As coincident detection occurs at the level of the synapse and astrocytic synaptic partners are highly heterogeneous [15], it is essential to study these processes locally. Currently, much of the data considers co-agonist regulation remotely, either in the bulk tissue or by measuring synaptic activity remotely at the neuronal soma. Understanding co-agonism at the subcellular level will be critical to understanding the physiological role(s) in circuit activity, learning and memory. The aim here is to review the current understanding of astrocyte-mediated co-agonism and explore its potential role in coincidence detection.

## 2. NMDARs

NMDA (*N*-methyl-d-aspartate) receptors (NMDAR) are ionotropic glutamate receptors that are ubiquitously expressed at excitatory synapses in the mammalian brain. NMDARs are heteromers with numerous subunit compositions. Currently, seven subunits are known, and these are classified into three subfamilies: GluN1, GluN2A–D, and GluN3A–B. Although only one GluN1 gene has been identified, eight splice-variants/isoforms exist (i.e., GluN1-1a–GluN1-4a and GluN1-1b–GluN1-b4) [16]. The subunits are homologous, and their tetrameric assembly forms a large variety of NMDAR subtypes. Functional tetrameric receptors require two GluN1 subunits and two non-GluN1 subunits (GluN2 or GluN3) and typically form di-heteromers of 2GluN1/2GluN2 or tri-heteromers of 2GluN1/GluN2/GluN3. This topic has been reviewed in detail by Paoletti et al. [16]. At least a dozen functionally distinct channels have been reported to date [16]. Their subunit composition, particularly GluN2 and GluN3, determines fundamental receptor properties that govern synaptic integration and plasticity, i.e., permeability to Ca^2+^, sensitivity to voltage-dependent block, channel kinetics, sensitivity to extracellular modulators, and dependence on co-agonist binding [17,18].

According to the properties conveyed by the specific subunits, glutamate binding is typically insufficient to open the NMDAR channel, which may be blocked by Mg^2+^ and requires depolarization as an expeller. Thus, the dual requirement enables NMDARs to act as a coincidence detector of simultaneous pre-synaptic (glutamate) and post-synaptic (depolarization) activities. In addition, the subunit composition also determines the receptor ligand; specifically, the GluN1 and GluN3 subunits bind glycine/d-serine, whereas GluN2 subunits bind glutamate [19]. Hence, for the classical NMDARs, GluN1/GluN2 receptors, the binding of two glutamates and two glycine/d-serine molecules are essential to open the channel [20]. In contrast, the GluD and GluN1/GluN3 receptors are gated purely by the co-agonist [21,22,23].

As the endogenous levels of extracellular d-serine and glycine are high (2–10 µM) [24,25,26,27,28,29], in comparison to the co-agonist site affinity (nanomolar range) [16], the co-agonist site was initially believed to be saturated [5] and therefore to have no bearing on NMDAR function. Nevertheless, it turned out that this assumption was wrong [13,30,31,32,33,34,35,36,37]. Furthermore, changes in the glycine site occupancy, in vitro [7,11,38] and in vivo [34], regulated synaptic NMDAR activity. Consequently, mechanisms regulating co-agonist availability are likely to regulate NMDAR function, synaptic plasticity, learning and memory.

In parallel to these findings was the realization that the co-agonists, d-serine and potentially glycine, are produced, stored, and released by the non-neuronal glial cells. In the context of the current review, the so-called co-agonism of NMDARs is of central interest. Co-agonism of glutamate and glycine at GluN1/GluN2 NMDARs enables inhibitory neurons [39] and glia [40,41,42] to engage in crosstalk with excitatory synapses. In the following section, focusing on glia, we will explore the molecular mechanisms underlying co-agonist availability at the synapse.

## 3. Molecular Mechanisms of Astrocyte Mediated Co-Agonism

The brain contains an equal number of neuronal and non-neuronal cells, including glia [43]. Among the glia, astrocytes influence diverse functions, including neurometabolic coupling, glutamate/glutamine cycle, synaptic transmission and synaptic plasticity. Astrocytes regulate synapse function by releasing neuro-active substances termed gliotransmitters (by analogy to neurotransmitters) [44,45]—e.g., d-serine, an endogenous NMDAR co-agonist. Mounting evidence indicates that glycine, another NMDAR co-agonist, is also released. In addition to co-agonist release, astrocytes regulate co-agonism by playing a central role in the d-serine lifecycle. Therefore, to understand the functional consequences of astrocyte in co-agonism, it is essential to unravel the molecular mechanisms of co-agonism initiation and termination at the synapse. To this end, the following section reviews the current understanding of molecular mechanisms underlying co-agonist availability at the synapse—i.e., co-agonist synthesis, release, uptake, and degradation.

### 3.1. Co-Agonist Synthesis

Glycine and d-serine concentrations are relatively high in the brain (2–10 µM) [25,26,27,28,29,46], where they are synthesized locally [47] from l-serine [28,48,49,50]. l-serine itself is synthesized from glucose by the phosphorylated pathway occurring exclusively in glia cells and predominantly astrocytes [48,51,52]. l-Serine is converted into glycine by serine/glycine hydroxymethyltransferase (sgHMT) [53] and d-serine by serine racemase (SR) [54,55,56,57,58] (Figure 1). Interestingly, the knock-out of SR reduced d-serine in the hippocampus and cerebral cortex by 80 to 90% [59,60,61,62]. Although l-serine is the primary precursor of d-serine [47], residual production in the *SR^-/-^* brain suggests an additional route for d-serine synthesis, albeit minor, possibly involving glycine cleavage system [63] or phosphoserine phosphatase [46]. These routes may have more importance in regions with low SR expression, e.g., the cerebellum [59,60].

SR initially appeared to be an astrocytic enzyme [10,54,64,65,66,67,68,69]. However, the cellular location of SR and thus d-serine production was recently debated [70,71]. In addition to d-serine production in astrocytes, it was proposed that l-serine is shuttled to neurons for racemization [72]. The debate primarily arose because the detection of SR in situ is sensitive to the antibodies and fixation conditions used [54,67]. In various conditions, SR was located to only astrocytes [8,54,69,73,74,75], astrocytes and neurons [76], neurons and oligodendrocytes [77], or only principal neurons [67,78,79].

As brain SR protein has a single molecular-weight band and one predicted amino acid sequence [54,57], it is generally accepted that there is only one isoform of SR in the brain. However, Neidle and Dunlop [80] reported two isoforms of mouse brain SR with similar molecular weight and enzymatic properties but structural differences. Since then, multiple transcripts have been reported [74,81]. The possible existence of astrocyte and neuronal-specific SR isoforms could explain the differential detection of neuronal and glial SR by distinct antibodies; isoform differences may also explain the differential regulation of SR in neurons and astrocytes [73,82]. Recently, SR distribution was examined without SR antibodies, using a transgenic mouse with GFP expression under the control of an SR promoter; this revealed that the SR promoter is active in both neurons and astrocytes [83].

Although present in neurons and astrocytes, the role of SR at these two locations is unresolved. It is intriguing to consider that their respective role could be distinct. Conditional KOs are promising tools to resolve this. However, so far, this technology has provided inconclusive data. While selective acute inhibition of d-serine synthesis or release by astrocytes reduced NMDAR function and impaired LTP [8,9,10,14,84,85], the conditional KO in astrocytes did not impair LTP [78]. Due to concerns over the time allowed for KO in astrocytes [70] and the poor recombination of endogenous floxed loci using hGFAP-CreER^T2^ mouse line [86], it would be interesting to use a different approach for the targeted expression of Cre in astrocytes—e.g., AAV. In contrast, depleting SR or l-serine in neurons impaired LTP [78]. However, in this case, the LTP impairment was inconsistently accompanied by changes in harmony with the co-agonism of synaptic NMDARs—i.e., reduced activity of synaptic NMDARs, reduced extracellular d-serine, and reduced NMDAR co-agonist site occupancy [50,78,87]. Notably, chronic depletion of SR (>45 days) in the post-synaptic neuron did not change the synaptic content of d-serine but, over time, altered the synaptic content of GluN2B [87]. As d-serine can alter the mobility of GluN2B in the membrane [88], this may indicate that neurons regulate extracellular d-serine at a site apart from the synaptic cleft, potentially controlling the subunit composition of synaptic NMDARs in adult CA1 pyramidal neurons. These findings suggest that neuronal l-serine and SR are critical for LTP but may not involve the co-agonism of synaptic NMDARs. For example, SR may be required for d-serine degradation and the production of pyruvate [65,76,89,90], and l-serine is needed for protein synthesis [91] and the production of phospholipids and sphingolipids [48,92,93]. Notably, according to the enzyme affinities for l-serine, these pathways are likely favored over SR [54,89].

The half-life of SR (4.5 h) [94] is much shorter than that of d-serine (12 h) [47]; consequently, the activity of SR may have a crucial role in regulating d-serine release. SR activity is regulated by numerous players, including ephrinB3 [95], glycolysis [96], DISC1 [73], Golga3 [94], GRIP [97], glycine [50,89,98] (also see [64]).

### 3.2. Release

Both astrocytes and neurons can release NMDAR co-agonists [99,100]. In the context of the current review, addressing the role of astrocytes in co-incident detection, we focus on co-agonists released by astrocytes and direct interested readers to other reviews exploring neuronal release [101,102]. Astrocytes can release d-serine under ambient or basal conditions [8,10,13,14,38] and in response to neuronal activity [9,10,11]. Both the tonic and active release depend on Ca^2+^ as they may be augmented by increasing extracellular Ca^2+^ and blocked by impairing Ca^2+^ release/influx or chelation of intracellular Ca^2+^ [10,14,66]. Multiple modes of d-serine release indicate several release mechanisms, and indeed, using primary culture, several mechanisms have been proposed. Here, we review the proposed mechanisms of co-agonist release by astrocytes.

#### 3.2.1. Exocytosis

Stimulation of non-NMDAR glutamate receptors on primary astrocytes, or the Ca^2+^ permeable ionotropic receptor ⍺7nAChR (α7 nicotinic acetylcholine receptor) in situ, triggers Ca^2+^- and SNARE-dependent vesicular release of d-serine by astrocytes [9,10,66,84,103,104]. Consistent with these findings, astrocytic d-serine concentrates within synaptic like micro-vesicles (SLMVs) in culture and in situ. Importantly, SLMVs are close to synapses, are accompanied by a significant intracellular Ca^2+^ store (endoplasmic reticulum) [105,106,107], and are equipped with proteins of the SNARE-dependent regulated secretory pathway, i.e., synaptic vesicle protein 2 (SV2), synaptobrevin (Sb2/VAMP2) and cellubrevin (VAMP3) [66,105,107,108,109,110].

Functional evidence for SNARE-dependent exocytosis of d-serine containing SLMVs rests on the inhibition of d-serine release using SNARE toxins (tetanus neurotoxin) and the exogenous expression of dominant-negative SNARE protein [9,10,66]. Nevertheless, it was astutely pointed out that this may also be explained by perturbed the SNARE-dependent membrane insertion of channels/transporters mediating d-serine release [45]. Of relevance to this point, Papouin et al. (2017) recently reported that tonic d-serine release is SNARE-independent and co-exists with SNARE-dependent release [70].

#### 3.2.2. Volume-Regulated Chloride/Anion Channel (VRAC)

VRACs are typically activated by cell swelling and provide a pathway for the out-flux of intracellular anions and amino acids, including glutamate, aspartate, and taurine [111]. In addition, VRAC mediates AMPA (⍺-amino-3-hydroxy-5-methyl-4-isoaxzolepropionic acid)-induced d-serine release from primary astrocytes [100]. The molecular identity of VRAC has been enigmatic; this is highlighted by its many names (volume-sensitive outwardly rectifying [Cl-] channel, VSOR [112]; volume-sensitive organic osmolyte[organic]-anion channel, VSOAC [113]; volume-sensitive chloride channels, I_Cl,vol_ [114]; and swelling activated Cl^-^ channel, I_Cl,swell_ [115]). The leucine-rich repeat-containing protein 8A (LRRC8A) appears to be an essential VRAC component [116,117] mediating receptor-activated amino acid release in astrocytes [113].

Although VRAC opening is classically Ca^2+^ independent, recent evidence indicates that Ca^2+^ regulates VRAC activity. Either receptor-induced Ca^2+^ increase can activate VRAC by an associated increase in cell volume [111,118], or independent of swelling, VRAC activity is modulated by G_q_-protein coupled receptors and downstream signalling cascades involving Ca^2+^-dependent protein kinases and ROS production [113,118,119,120,121,122,123]. As Ca^2+^ nano-domains modulate VRAC activity, VRAC-mediated d-serine release may preserve the input specificity of synaptic transmission [122,123]. While receptor-mediated Ca^2+^ transients are rapid, the modulation of VRAC current develops gradually over 15 to 20 min [121,122]. The gradual action favors intercellular communication on a long timescale, even when Ca^2+^ signals are transient and confined to fine subcellular structures.

#### 3.2.3. Hemichannels

Cx43 hemichannels mediate the release of glutamate, ATP, glutathione, and d-serine from astrocytes and glioma cells [85,124,125,126]. Hemichannels are open at intracellular Ca^2+^ in the range of 0.1 to 1 µM and peak at 0.5 µM [85,127,128], values typically observed under physiological conditions [129,130], and at membrane potentials typical of astrocytes [85,131]. Lowering extracellular Ca^2+^ increases the opening probability of Cx43, whereas 1 mM extracellular Ca^2+^ closes them [125,132]. In vitro studies indicate that hemichannels participate in activity-dependent d-serine release either directly, by opening in response to Ca^2+^ [85,127,133,134] or IP_3_ [135,136,137], or indirectly by facilitating Ca^2+^ entry [138], which subsequently activates other [Ca^2+^]_i_-dependent d-serine release mechanisms [85]. Nevertheless, the significance of hemichannels for gliotransmission in vivo is uncertain [139,140].

#### 3.2.4. Reverse Uptake

A transporter can potentially mediate substrate uptake and release, and its stoichiometry is a critical factor that controls the driving force and thus the transmitter flux direction. Thus, in principle, d-serine may also be released from astrocytes by reverse transport. To date, ‘alanine, serine, cysteine transporter 2′ (ASCT2) has been the leading candidate for the reverse uptake of d-serine by astrocytes in vitro; however, current data indicates that this probably does not occur in vivo. In situ ASCT2 is detected on neurons and retinal glia but not in astrocytes or Bergmann glia [141,142]. In addition, triggering amino acid hetero-exchange through ASCT2 transporters in vivo did not induce significant d-serine release [143]. Finally, ASCT2 transporters have a much higher affinity for l-glutamine and l-serine and, under physiological conditions, should preferentially release these molecules [72,143,144]. The contribution of astrocytic ASCT1 to d-serine release in situ is also unclear. Although ASCT1 KO reduced extracellular d-serine, indicating a role in d-serine release, the selective activation of ASCT1 hetero-exchange in acute slices did not elicit detectable release of endogenous d-serine or glycine [145]. One possible explanation is that ASCT1 reverse uptake is maximal under resting conditions.

Glycine, the other co-agonist of NMDARs, can be released from astrocytes via a non-vesicular mechanism such as reverse transport by the glycine transporter GlyT1 (reviewed by [102,146,147,148]. The glial glycine transporter, GlyT1b, has a stoichiometry of 2Na^+^/Cl^-^/glycine, which predicts that glycine can be exported or imported, depending on the physiological conditions [149,150]. Activation of reverse uptake occurs when there is an increase in intracellular glycine or intracellular Na^+^ and following membrane depolarization [149,150,151]. Thus reverse-uptake could occur in response to local increases in the intracellular Na^+^ concentrations resulting from either the activation of glia AMPA receptors [146,149,152,153,154] or enhanced Na^+^/Ca^2+^-exchanger (NCX) activity following a pure Ca^2+^ response. The release is also enhanced by lowering extracellular Ca^2+^ and K^+^ [150]. In astrocytes, receptor stimulation and activation of the G_q_PCR-PLC signalling cascade may also enhance glycine release via GlyT1 [155]. By estimation, reversed glycine uptake could increase extracellular glycine from a sub-saturating level (~100 nM) to the low micromolar range [146]. However, the functional significance of reverse transport in vivo is unknown. Some essential biophysical details are needed to clarify the role of reverse transport, e.g., the quantification of membrane depolarization and ion concentrations in subcellular astrocytic compartments and transmitter diffusibility. The employment of Na^+^ imaging and genetically encoded voltage sensors will provide some illumination in the future.

#### 3.2.5. P2X Purinoceptor 7

ATP activates P2X purinoceptor 7 (P2X7) receptors opening a large conductance pore, which can mediate the release of glutamate [156] and ATP [157,158]. In addition, P2X7could potentially release d-serine [45]. Notably, cytosolic Ca^2+^ does not regulate P2X7 channel gating.

#### 3.2.6. Astrocytic Mechanisms of Tonic and Active Release of Co-Agonists

Astrocytes participate in the tonic and active release of NMDAR co-agonists. Tonic release accounts for the ambient occupation of the NMDAR co-agonist site. Experimentally, this has been assessed by applying low-frequency stimulation (0.05 Hz) [10] and observing the synaptic NMDAR function in the absence and then presence of saturating exogenous co-agonist. In contrast, the active release of co-agonist has been triggered by high-frequency (50 Hz) trains of neuronal activity and observed by monitoring NMDAR function before and after conditioning stimuli [10]. However, because strong stimulation can trigger immediate active release under conditions of low-frequency stimulation [11,12], it is apparent that this experimental distinction may not be so clear. It may therefore be essential to control the level of synaptic stimulation to isolate tonic release experimentally.

Tonic and active release have distinct mechanisms but are both Ca^2+^-dependent [9,10,14] (Figure 2). As ASCT1 and P2X7 are Ca^2+^-insensitive, they are unlikely to play a role in either tonic or active release. Experimental data strongly indicates that exocytosis mediates the active release of d-serine [9,10]. In contrast, tonic release is SNARE-independent [9], regulated by Ca^2+^ influx channels (transient receptor potential ankyrin 1, TRPA1) and blocked by conditions that lower the resting Ca^2+^ level (100–150 nM) [129,130] to 50–80 nM [9,10,14]. The blocking of d-serine release at [Ca^2+^]_cyto_ <100 nM is consistent with the involvement of hemichannels [85,127]. However, it is not possible to rule out other Ca^2+^-sensitive mechanisms. Indeed, both VRAC activity [122] and d-serine release may be regulated by TRPC (transient receptor potential canonical)-mediated Ca^2+^ influx [159]. As commonly used inhibitors block hemichannel and VRAC [160,161], their importance in the tonic release is unclear. The recent cloning of the VRAC channel will hopefully provide the means to solve this issue [113].

Tonic and active release, as defined here, differ in their temporal characteristics. The tonic release of co-agonists by astrocytes sets the ambient level of co-agonism. Furthermore, it is expected that the regulation of tonic release will produce changes in NMDAR function that are slowly adapting and sustained over long periods. In this sense, tonic release modulates NMDAR function [44,162]. Such ambient co-agonism may play a role in supporting NMDAR function under basal conditions. Tonic release may also limit NMDAR activity by promoting NMDAR desensitization through “glycine-dependent desensitization” [163,164,165,166]. In contrast, the active release of co-agonists by astrocytes produces a transient enhancement of NMDAR function [10] and influences neuronal activity in real time. LTP induction at the Schaffer collateral-CA1 (cornu Ammonis subfield 1) synapses depends on the tonic (SNARE-independent, TRPA1-dependent) [9,14] and active co-agonism (SNARE-dependent) [9,10]. It is currently not clear why LTP requires active and tonic co-agonism. The simplest explanation is that active co-agonism has an insufficient capacity to saturate the NMDAR glycine site (we estimated that SLMV contains approximately 120 molecules of d-serine [105]). Thus, tonic co-agonism provides a necessary leg up for active release to exceed saturating concentrations of co-agonist at the synapse (>1.5 µM [152]).

### 3.3. Termination of Co-Agonism

Synaptic NMDARs are not saturated, indicating that active transport mechanisms reduce d-serine and glycine in the synaptic cleft. Uptake is also important for the termination of co-agonism and the availability of d-serine and glycine for future release. As d-serine is an agonist of the strychnine-insensitive glycine binding site on the NMDAR, three- to fourfold more potent than the co-agonist glycine [29], clearance of d-serine is likely to have profound physiological consequences.

#### 3.3.1. Glycine Uptake

At excitatory synapses, glycine transporters maintain glycine below saturation of NMDARs [7,33]. Many glycine transporters have been cloned in the mammalian CNS, and all are derived from two genes: GlyT1 (GlyT1a–GlyT1c) and GlyT2 (GlyT2a–GlyT2b). GlyT1 is located in fine astrocyte processes around glycinergic and non-glycinergic neurons and areas devoid of strychnine-sensitive receptors [167,168]. Glycine is rapidly accumulated into presynaptic terminals by GlyT2a, whereas GlyT1b controls the extracellular glycine concentration [35,149,169,170,171]. GlyT1 has, in principle, the ionic strength/accumulative power needed to reduce [gly]_o_ well below values that saturate the glycine site of NMDAR (40 nM to 1 µM [29,152]).

The proline transporter, SLC6A20A [172], is a novel transporter of glycine in the brain that regulates extracellular glycine and NMDAR function [173]. SLC6A20A cotransport glycine with Na^+^ and Cl^-^ ions. SLC6A20A is expressed in glia (astrocytes and microglia) and is less prominent in neurons [173]. Unlike GlyT1 and GlyT2, which are more strongly expressed in the brain stem [40,174,175], SLC6A20A are located in various brain regions, including the hippocampus and cortex [173].

#### 3.3.2. D-serine Uptake

Studies employing synaptosomes and competitive inhibitors have identified the following d-serine transporters: the Na^+^-independent transporter asc-1 [142,176,177,178] and the Na^+^-dependent transporters ASCT1 [145], ASCT2 [178], and System A transporters (SAT1 and SAT2) [176]. The immunohistochemistry indicates that, within the CNS, Asc-1 is a neuronal protein concentrated in the presynaptic complexes [177,179,180,181]; ASCT2 is limited to neuronal dendrites and, except for retinal glia, is not found in glia [142]; ASCT1 is present on astrocytes [142,145]; SAT1 is expressed mainly in GABAergic neurons [176,182,183,184,185,186,187,188,189], pyramidal neurons lack appreciable immunoreactivity [176,183,184,190], and SAT2 is expressed in the somatodendritic compartments of glutamatergic neurons, barely detected in interneurons, and potentially glia [185,189,191,192,193,194,195]. Although helpful in identifying transporters, synaptosomes and inhibitors, which are themselves substrates for transport, have limited use for studying d-serine regulation in vivo [196]. In order to understand the physiological importance of these transporters, we have chosen to focus, when possible, on studies using intact preparations and KOs or inhibitors that are not also substrates for transport. Measurements of extracellular d-serine either in vivo or in acute brain slices indicate that, overall, asc-1 and ASCT1/2 either release d-serine or are unimportant [145,180,196,197]. So far, only System A appears to accumulate intracellular d-serine in vivo. However, this relied on inhibition by the System A substrate MeAIB [176], which is also a substrate of the novel glycine transporter, SLC6A20A [173]; as such, confirmation awaits the development of a specific non-competitive inhibitor.

The empirically determined transporter roles appear at odds with their biophysical properties. As the external d-serine is low, 5–8 µM [26,198], the high affinity of asc-1 for d-serine (19 µM) and rapid uptake [178] have made it the primary candidate for d-serine transport in vivo. In contrast, the low affinity of System A (2.3 mM) [176], ASCT1 (150 µM) [199] and ASCT2 (110–700 µM) [142,199,200,201,202] has indicated that they are unlikely to contribute significantly to d-serine uptake in vivo [178]. A more detailed study of the spatial–temporal uptake of extracellular d-serine may provide clarity. From the published data, it is apparent that synaptic d-serine may be regulated differentially from bulk extracellular d-serine. Briefly, inhibition of asc-1 and System A, respectively, increased and decreased synaptic NMDAR function [176,197], indicating that, in contrast to bulk extracellular d-serine measurements, at the synaptic level, asc-1 takes up d-serine, and System A releases d-serine. Although blocking asc-1 only modestly enhanced synaptic NMDAR function, this is consistent with the observation that the sub-saturated co-agonist sites are held close to saturation [9,10]. Extracellular d-serine has several proposed functions, including co-agonism of synaptic NMDARs [203], regulating trafficking of extra-synaptic NMDARs [88], and a pool for shuttling from glutamatergic to GABAergic neurons [79]. Given these distinct roles and the differential regulation of uptake, spatially distinct pools of extracellular d-serine may exist and be served by different pools of transporters (Figure 3). It is compelling that asc-1, operating close to equilibrium, might release d-serine at rest to maintain sub-saturating levels of co-agonist and transiently flip to d-serine uptake to clear synaptic d-serine following transient co-agonism. Furthermore, a concentration of extracellular d-serine at non-synaptic sites could explain how the low-affinity System A transporter can take up d-serine in vivo. A detailed analysis of the spatial and temporal profiling of d-serine pools and uptake is essential to our understanding. Such studies may benefit from fluorescent d-serine biosensors under development [204].

### 3.4. Recycling of d-serine and Glycine

Brain d-serine has a half-life of approximately 16 h [47,80], but its degradative pathway is unresolved. In the mammalian brain, d-serine may be degraded by d-amino acid oxidase (DAAO) and SR [54].

DAAO catalyzes the oxidative deamination of neutral d-amino acids. DAAO is low in the higher brain areas/forebrain and occurs primarily in the brain stem, cerebellum, and spinal cord, concentrated in astrocytes of the hindbrain and cerebellum [68,205,206]. DAAO may participate in the catabolism of d-serine in the cerebellum and medulla oblongata. There appear to be other mechanisms for catabolism of endogenous d-serine in the rostral brain [68,207,208]. Inhibitors of DAAO can elevate endogenous d-serine levels in lower brain regions, but to a much smaller degree in the forebrain, e.g., the hippocampus and cerebral cortex [209]. Furthermore, d-serine levels were normal in the cerebrum, hippocampus, hypothalamus, pituitary gland, and pineal gland of mutant mice lacking DAAO (*ddY/DAAO^–^*) [208]. At odds with this, NMDAR-dependent learning and behaviour, hippocampal LTP, and NMDAR function were all enhanced in mutant mice lacking DAAO activity [210,211]. These findings suggest that DAAO plays a role in regulating the very local synaptic environment. Nevertheless, because the potent NMDAR co-agonist, d-alanine [18], is increased fivefold in the hippocampus of *ddY/DAAO^–^* mice [208], the role of DAAO in d-serine catabolism requires clarification. Heightened anxiety may also have played a role [212].

SR is a unique pyridoxal 5′-phosphate enzyme present in astrocytes and neurons (see Section 3.1). Its enzymatic activity is dependent on divalent cations, alkaline pH, and ATP [58,80,90] and can remove d/l-serine by β-elimination [58,65,80,89] or racemization. d-serine is removed directly by β-elimination [65,89,90] or by racemization and subsequent β-elimination [89]. Given the relative racemase and β-elimination activities of SR, degradation of d-serine is only expected if its concentration approaches or exceeds that of l-serine [89]. Indeed, appreciable d-serine degradation was only observed in vitro when l-serine dropped to 0.1–0.2 mM [65] and in the presence of supra-physiological d-serine (5 mM) [65]. Hence, because astrocytes continuously synthesize l-serine (1 mM) and cytosolic d-serine is relatively low (0.3 mM) [65,152,213,214], SR is unlikely to remove d-serine in astrocytes. In contrast, the higher d-serine and lower l-serine expected in neurons may favor d-serine degradation by SR. d-serine may be stored in astrocytic SLMV and extracellular space to avoid the risk of accumulating cytosolic d-serine [65].

## 4. Detection of Coincident Neuronal and Astrocytic Activities by NMDARs

Although astrocytes can regulate NMDAR function, a role in coincidence detection has seemed unlikely as it requires participation in generating associations between temporally and spatially linked events. Astrocyte biology has long held that astrocytes operate on a much slower time scale than neurons, emphasizing volume transmission over point-to-point communication [162]. The disconnect of astrocyte activity from the immediate spatial and temporal activity of neuronal networks led to the idea that astrocytes, rather than participating in the ongoing activity, are regulators of synaptic transmission and plasticity. In line with this view, although NMDAR activation depends on glutamate and co-agonist binding, only glutamate is believed to participate as a transmitter—i.e., released in an activity-dependent manner. In contrast, the co-agonists glycine and d-serine are present at more constant levels, indicating a modulatory function [215]. Interestingly, as discussed in Section 3.2.6, the presence of both tonic and activity-dependent release of d-serine could play distinct roles, as has been proposed recently [9,216].

Aside from the acknowledged role of astrocytes in synaptic regulation, snippets of experimental evidence have begun to suggest that astrocytes might also be capable of actively participating in synaptic transmission and coincidence detection: (1) astrocytes are capable of generating local Ca^2+^ signals at the synaptic level [15,217,218] and may therefore preserve and participate in point-to-point communication; (2) initially thought to take seconds to respond, and therefore too slow to actively participate in synaptic transmission, recent studies show that astrocytes are capable of responding within hundreds and even tens of milliseconds in vitro [217,219] and in vivo [220,221]; (3) in comparison with previous studies examining the occupancy of the glycine site under basal conditions—tonic/ambient co-agonist availability—the NMDAR glycine site occupancy can be actively regulated by astrocytes in the hippocampus and amygdala [9,10,11,12]; (4) critically, actively released co-agonists had an immediate impact on evoked responses, indicating a near-instantaneous release of co-agonists in response to presynaptic stimuli [11,12].

### 4.1. Astrocytes Detect Activity at Segregated Synapse

The points above allow for the active participation of astrocytes in point-to-point synaptic transmission and plasticity. However, simply adding astrocyte-mediated co-agonism to the classical co-incident detection scheme would make astrocytes mere relays for synaptic transmission. Assuming astrocyte-mediated co-agonism is essential, how might astrocytes’ unique structure and functional properties enrich coincidence detection?

In addition to participating in synaptic transmission, two landmark studies by Li et al. (2009, 2013) [11,12] demonstrated that the amount of co-agonist released by astrocytes scales with the level and pattern of synaptic activity. These findings strongly suggest that, as with neurons, astrocytes perform spatial–temporal integration of synaptic activity. While astrocyte integration is not a new concept [44], the distinction made by Li et al. is that astrocytes responded rapidly and participated in synaptic transmission. To recapitulate this important point, integration of converging neuronal signals by astrocytes translates into a coordinated increase of NMDAR activity through co-agonist release.

To understand how spatial–temporal integration performed by astrocytes might differ from that of neurons, we next consider astrocyte morphology. Astrocyte morphology, which plays a vital role in astrocyte physiology, is striking next to neurons. If astrocytes are involved in coincidence detection and synaptic integration, their functional anatomy will undoubtedly play an important role. Each astrocyte occupies an exclusive ‘territory’ or ‘domain’ [222,223], containing more than 100,000 synapses [222]. Astrocyte’s unique reticular morphology forms bridges between synapses irrespective of their pre-synaptic and post-synaptic components [15,224] (Figure 4). Thus, astrocytes potentially provide a conduit for communication between related and unrelated synapses. In the context of the spatial–temporal integration described above, this could provide conditions for the coordinated increases in the activation of NMDAR at closely located but unrelated synapses—segregated synapses. In contrast to neuronal dendritic integration, which performs spatial–temporal integration of synaptic events in a segregated group of inputs converging on a single hippocampal neuron, the astrocyte would be able to detect and integrate the synaptic signal at segregated synapses [11,44]. Activity-dependent co-agonism by astrocytes would promote plasticity in clusters of unrelated but co-active synapses (Figure 4). As a single astrocyte serves 300 to 600 dendrites [223], these principles of synaptic integration may extend to dendrites.

In addition to the spatial–temporal integration described above, the astrocytes’ structural hierarchy might provide further opportunities for compartmentalization and integration. The astrocytes’ reticular structure emanates from three to five major processes, which are themselves connected through the somatic compartment. It is possible that the major processes divide an astrocytes territory, and the synapses it serves, into functional districts and gates their interaction. Indeed, from Ca^2+^ imaging studies, it is known that major processes can operate independently or together. Accordingly, a single astrocyte may integrate synaptic information and release the co-agonist at a single synapse [15,217,218,225] or clusters of synapses of varying sizes. Synapse clusters may include the territory of a single process, several processes, or the whole astrocyte.

Although synapse independence increases the computational power of neuronal networks [226], there is a need for cellular mechanisms mediating synchronous plastic modifications in a group of co-active synapses. Co-agonism by astrocytes could optimize the conditions for encoding and retaining specific information about different network activity patterns. This function is distinct from the afferent activity patterns stored by dendritic (synapse) clusters. To this end, the maximal activation of NMDARs at segregated synapses, soaked in the co-agonist released in an activity-dependent fashion, would permit the induction of NMDAR-dependent synaptic plasticity at depolarized membrane potentials while retaining synapse functional independence at the resting membrane potential [11].

### 4.2. Differential Shaping of the Coincidence Window by Glycine and d-serine

The level of NMDA receptor glycine site occupancy under steady-state conditions or low activity is synapse-dependent [8,10,13,103,104], indicating a circuit-dependent role for the glycine site. In addition, the identity of the co-agonist, glycine or d-serine, is synapse-specific [8,12,227,228], developmentally regulated [88,228], and possibly activity-dependent [9,12,145,228]. These patterns are tied to the NMDAR subtypes expression pattern at specific synapses [88,228]. Nevertheless, with the current level of understanding, it is hard to appreciate why a given synapse might benefit from either tuning the glycine site saturation or opting for one co-agonist over the other. The level of co-agonist site saturation may be tuned (between unsaturated to saturated) to turn coincident detection of segregated inputs on and off. In some instances, it has been suggested that neurons provide d-serine and glycine [38]. The participation of neurons in co-agonism may represent situations where the circuit does not require or desire the strengthening of synapses based on the coincident activation of segregated input but wishes to retain activity-dependent co-agonism.

Regarding the heterogeneity of the co-agonist identity of glycine or d-serine, one possibility could be to regulate the subtype of subunit composing synaptic NMDARs [88]. Another possibility could be that the choice of two co-agonists could provide a means to shape the plasticity rules governing coincident activity in segregated inputs. In this framework, the lifetime of the co-agonist in the synapse dictates the window for coincident activities. Moreover, the differential regulation of d-serine and glycine could provide a means to fine-tune (stretch or compress) this window to the specific needs of a given circuit.

## 5. Conclusions

Co-agonism of NMDARs by astrocytes has attracted much interest over the last two decades, its physiological relevance and even existence being heavily debated. In recent years, a surge in studies reporting a neuronal role in co-agonism has at times appeared to challenge the role of astrocytes. However, on balance of the evidence provided above, astrocytic co-agonism remains sound. The emergence of neurons in co-agonism, rather than supplanting an astrocytic role, acts to enrich it. Astrocytes have not been considered capable of co-incident detection. The current review aimed to reassess this point of view. Many of the reservations concerning the properties of astrocytic activity that may present a barrier to co-incidence detection have been softened or removed in recent years.

Furthermore, landmark studies by Li et al. [11,12] have observed activity-dependent co-agonism and proposed a novel role for co-agonism for the coincidence detection of co-active segregated synapses. At this time, the question of co-agonism and co-incident detection is intriguing and remains open. The proposed coincident detection of co-active synapses will be heavily influenced by the spatial–temporal properties of co-agonism and the physical arrangement of synapses. Therefore, it will be necessary to incorporate the dendritic and reticular structures of neurons and astrocytes, respectively. The sub-cellular anatomy is essential to account for the heterogeneity of tripartite synapses and the extracellular space between synapses and astrocytic morphology. As co-agonism is regulated by co-agonist uptake and metabolism, in the future, these processes should also be studied in relation to the subcellular structure and synaptic microenvironment. The fine mechanistic and biophysical details are critical to understanding the physiological role of co-agonism and inform more realistic neuronal models, incorporating highly consequential and dynamic co-agonism by astrocytes.

## Figures and Tables

**Figure 1 ijms-22-07258-f001:**
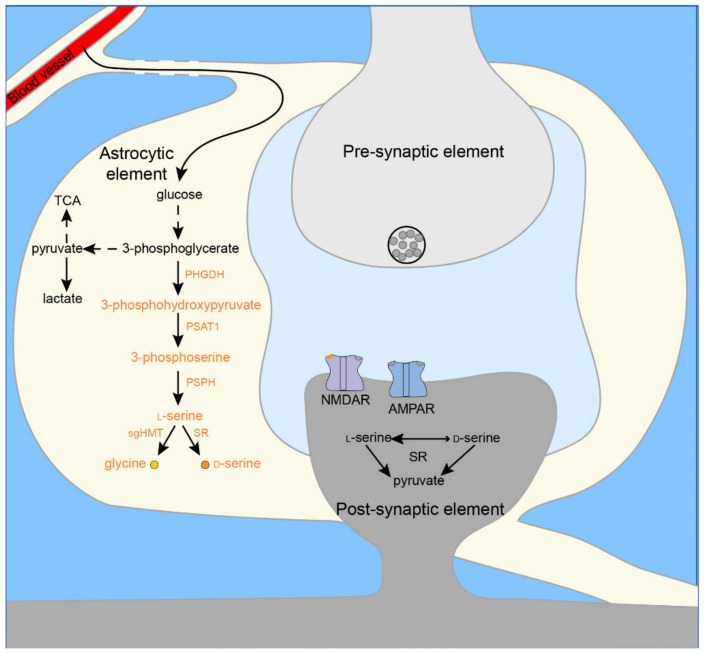
Scheme of l-serine, d-serine, and glycine synthesis in the central nervous system. In astrocytes, l-serine is synthesized from glucose by the phosphorylated pathway (orange) branching from glycolysis (black). 3-phosphoglycerate, an intermediate of glycolysis, is converted to 3-phosphohydroxypyruvate by 3-phosphoglycerate dehydrogenase (PHGDH). Subsequently, 3-phosphohydroxypyruvate is metabolized into 3-phosphoserine by phosphohydroxypyruvate aminotransferase (PSAT1). Finally, 3-phosphoserine is hydrolyzed to l-serine by phosphoserine phosphatase (PSPH). l-Serine is converted into glycine by serine/glycine hydroxymethyltransferase (sgHMT) and d-serine by serine racemase (SR). In neurons, conditions may favor the conversion of d-serine into l-serine and pyruvate. In addition, l-Serine may be an important source of sphingolipids and phospholipids. Abbreviations: TCA, tricarboxylic acid cycle; PHGDH, 3-phosphoglycerate dehydrogenase; PSAT1, phosphohydroxypyruvate aminotransferase; PSPH, phosphoserine phosphatase; sgHMT, serine/glycine hydroxymethyltransferase; SR, serine racemase; NMDAR, *N*-methyl-d-aspartate receptors; AMPAR, ⍺-amino-3-hydroxy-5-methyl-4-isoaxzolepropionic acid receptor.

**Figure 2 ijms-22-07258-f002:**
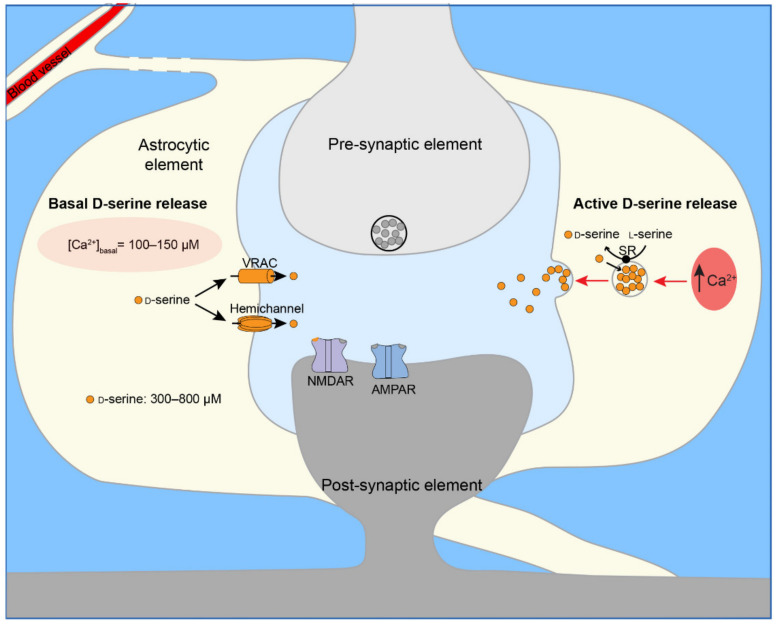
Scheme of a glutamatergic synapse showing the mechanisms of Ca^2+^-dependent d-serine release by astrocytes. In astrocytes, d-serine is present in the cytosol (300–800 µM) and concentrated within synaptic-like micro-vesicles (6 mM). Vesicular uptake of d-serine is mediated by an unidentified vesicular d-serine transporter and facilitated by vesicle-associated serine racemase activity (SR). (**Left**) Under basal conditions, VRAC (volume regulated anion channel) and hemichannels release cytosolic d-serine. Basal release is regulated by [Ca^2+^]_basal_. (**Right**) High-frequency afferent stimulation (50 Hz) triggers Ca^2+^- and SNARE-dependent vesicular release of d-serine by astrocytes. Following d-serine release into the synaptic cleft, d-serine binds to synaptic NMDAR containing GluN2A. Abbreviations: SR, serine racemase; VRAC, volume regulated anion channel; NMDAR, *N*-methyl-d-aspartate receptors; AMPAR, α-amino-3-hydroxy-5-methyl-4-isoaxzolepropionic acid receptor.

**Figure 3 ijms-22-07258-f003:**
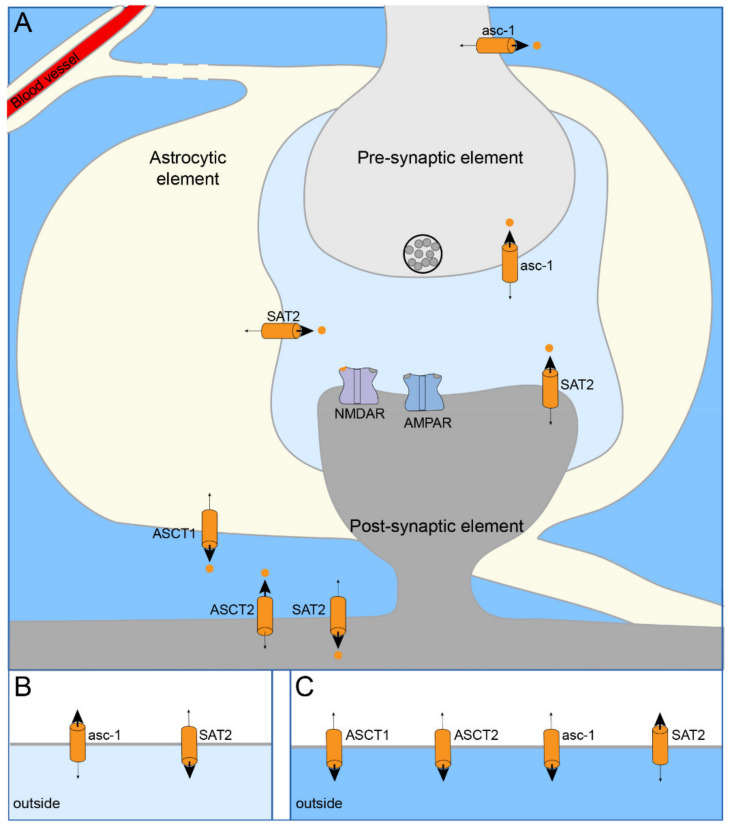
Scheme showing d-serine uptake and reverse uptake at glutamatergic synapses. (**A**) While astrocytes express ASCT1 and SAT2, post-synaptic neurons express ASCT2 and SAT2, and pre-synaptic neurons express asc-1. (**B**) At the synapse, asc-1 mediates d-serine uptake, and SAT2 mediates reverse uptake. (**C**) These roles are reversed outside of the synapse; SAT2 mediates uptake, and asc-1 mediates d-serine reverse uptake. ASCT1/2 also mediate reverse uptake outside of the synapse. Abbreviations: SAT2, System A transporter 2; ASCT1/2, Na^+^-dependent alanine-serine-cysteine transporter 1/2; asc-1, Na^+^-independent alanine-serine-cysteine transporter 1; NMDAR, *N*-methyl-d-aspartate receptors; AMPAR, α-amino-3-hydroxy-5-methyl-4-isoaxzolepropionic acid receptor.

**Figure 4 ijms-22-07258-f004:**
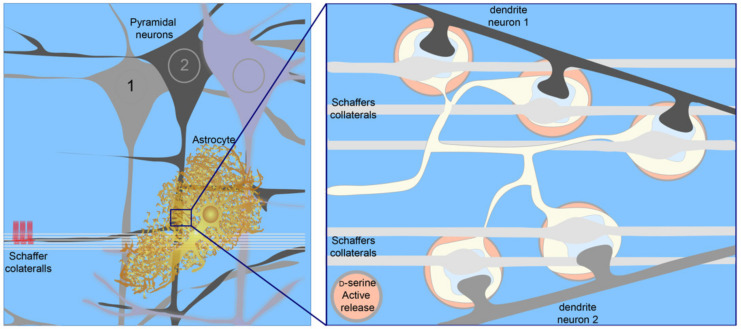
Schematic showing astrocyte interaction with related synapses (located on the same post-synaptic neuron) and segregated synapses (located on different post-synaptic neurons). Astrocytes detect coincident activity at related and segregated synapses, perform spatial–temporal integration of the activity patterns, and subsequently release an NMDAR co-agonist, i.e., d-serine or glycine. The amount of co-agonist released scales with the level and pattern of synaptic activity. Thus, activity-dependent co-agonism promotes plasticity in clusters of co-active synapses, regardless of their relationship status.

## Data Availability

No new data were created or analyzed in this study. Data sharing is not applicable to this article.

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
