# Peer review of "NMDARs, Coincidence Detectors of Astrocytic and Neuronal Activities"

_ijms, 2021, doi:10.3390/ijms22147258_

Round 1

Reviewer 1 Report

The review by Sherwood, Oliet, and Panatier thoroughly summarizes the current knowledge about co-agonist activities on NMDARs. The impact of the glial function on synaptic activity is pointed out, focusing on the tuning of coincident detection made by NMDARs. The main body of the text deals with the nature of the co-agonists (D-serine and glycine), their synthesis, release, and reuptake pathways. The last part of the review eventually addresses the impact of the glial role on NMDARs co-agonism in coordinating the activity of segregated synapses, which I believe is of particular interest.

Main comments:

The topic of this review is of particular relevance, and the authors did a good job in reporting a considerable portion of the relevant literature. Nevertheless, few adjustments are needed to improve the manuscript.

1) An extensive English revision is needed (e.g., for grammar, typos, use of punctuation, syntax). In my opinion, this is the main fault of this text in its current form.

2) The definition of Figure 1 is poor.

3) Considering the length of the text and the abundance of different sections, an “index” part should be added at the beginning, if possible. It is sometimes difficult to “navigate” in the different sections and keep an eye on the general structure of the review.

4) I agree with the authors (line 535) that the main message in this review is to re-assess the point-of-view of the glial function. The glial role in transient (point-to-point) synaptic activity needs to be explored further and indeed goes beyond the tripartite synapse, including the possibility of these cells to influence the activity of “clusters of unrelated but co-active synapses” (line 495). The concept of “coincident activation of segregated inputs” (line 519) is of particular relevance for this review and the scientific community in general to evolve the concept of glial “support” of neuronal activity. In this perspective, the review will benefit from an additional figure, reporting a glial cell interaction scheme with different synapses (on the same neuron and different neighboring neurons, summarizing sections 4.1 and 4.2).

Author Response

We would like to thank the reviewer for his thorough evaluation of our manuscript and his constructive and pertinent comments.  As you will see from our reply below, we have tried to answer as much as possible all comments and questions by doing additional and more detailed figures.

1) An extensive English revision is needed (e.g., for grammar, typos, use of punctuation, syntax). In my opinion, this is the main fault of this text in its current form.

Thank you for your comments. A native English speaker has extensively modified the English. Correcting for grammar, spelling, punctuation, and syntax. In addition, complex sentences were simplified to improve readability. We also took this opportunity to ensure that where appropriate the text was in the active form. We hope that the revised text is adequately improved and look forward to the reviewer’s comments and further suggestion.

2) The definition of Figure 1 is poor.

We have worked on improving the figure legends and hope these are satisfactory.

3) Considering the length of the text and the abundance of different sections, an “index” part should be added at the beginning, if possible. It is sometimes difficult to “navigate” in the different sections and keep an eye on the general structure of the review.

Thank you for the suggestion an index has been added (line 24).

4) I agree with the authors (line 535) that the main message in this review is to re-assess the point-of-view of the glial function. The glial role in transient (point-to-point) synaptic activity needs to be explored further and indeed goes beyond the tripartite synapse, including the possibility of these cells to influence the activity of “clusters of unrelated but co-active synapses” (line 495). The concept of “coincident activation of segregated inputs” (line 519) is of particular relevance for this review and the scientific community in general to evolve the concept of glial “support” of neuronal activity. In this perspective, the review will benefit from an additional figure, reporting a glial cell interaction scheme with different synapses (on the same neuron and different neighboring neurons, summarizing sections 4.1 and 4.2).

Thank you for this suggestion that will improve the visibility and clarity of the manuscript. The suggested figure has been added (Figure 4).

Reviewer 2 Report

Accumulated studies reveal the multiple functions of astrocytes. This review article aims to review the current knowledge about how astrocyte-mediate co-agonism and explore its role in NMDA receptor coincidence detection.

The article is well written with clear outlines and enriched information.

I only have few minor comments.

  1. It will be helpful to include a simplified figure for L-serine, D-serine, glycine metabolism.
  2. The figure is blurring; it needs a better resolution.
  3. The right panel of Figure 1 should also label the astrocyte terminal, pre-and post- synapse.
  4. Please include that ASCT is an acronym standing for alanine, serein, cysteine transporter2 in text.
  5. Does NCX state sodium-calcium exchanger (line 264).
  6. Please describe the full name of TRPC (line 298).
  7. Please provide the full name of SC (line 320).
  8. The full name of SR is mentioned twice (line 422).

Author Response

We would like to thank the reviewer for his thorough evaluation of our manuscript and his constructive and pertinent comments.  As you will see from our reply below, we have tried to answer as much as possible all comments and questions by doing additional and more detailed figures.

  1. It will be helpful to include a simplified figure for L-serine, D-serine, glycine metabolism.
    A simplified figure has been added (Figure 1).

  1. The figure is blurring; it needs a better resolution.
    We have improved the resolution of the figure to 300dpi.

  1. The right panel of Figure 1 should also label the astrocyte terminal, pre-and post- synapse.
    We have added these definitions to all figures.

  1. Please include that ASCT is an acronym standing for alanine, serein, cysteine transporter2 in text.
    The requested definition was added to the main text (line 264, word track mode = no markup).

  1. Does NCX state sodium-calcium exchanger (line 264).
    The requested definition was added to the main text (line 284).

  1. Please describe the full name of TRPC (line 298).
    The requested definition was added to the main text (line 320).

  1. Please provide the full name of SC (line 320).
    The requested definition was added to the main text (line 346).

  1. The full name of SR is mentioned twice (line 422).
    The duplicated definition of SR was removed.
